# Integrating niche and occupancy models to infer the distribution of an endemic fossorial snake (*Atractus lasallei*)

**Camilo Alejandro Cruz-Arroyave**[1], **Felipe A. Toro-Cardona**[2], **Juan Luis Parra**[2]*

**1** Grupo Herpetológico de Antioquia, Instituto de Biología, Universidad de Antioquia, Medellín, Colombia,
**2** Grupo de Ecología y Evolución de Vertebrados, Instituto de Biología, Universidad de Antioquia, Medellín, Colombia

* juanl.parra@udea.edu.co

**Data Availability Statement:** The data underlying the results presented in the study are available from DOI: 10.5061/dryad.5qfttdzff. Information about collected material can be found in https://ipt.

## Abstract

Understanding species distribution and habitat preferences is crucial for effective conservation strategies. However, the lack of information about population responses to environmental change at different scales hinders effective conservation measures. In this study, we estimate the potential and realized distribution of *Atractus lasallei*, a semi-fossorial snake endemic to the northwestern region of Colombia. We modelled the potential distribution of *A. lasallei* based on ecological niche theory (using maxent), and habitat use was characterized while accounting for imperfect detection using a single-season occupancy model. Our results suggest that *A. lasallei* selects areas characterized by slopes below 10˚, with high average annual precipitation (>2500mm/year) and herbaceous and shrubby vegetation. Its potential distribution encompasses the northern Central Cordillera and two smaller centers along the Western Cordillera, but its habitat is heavily fragmented within this potential distribution. When the two models are combined, the species' realized distribution sums up to 935 km², highlighting its vulnerability. We recommend approaches that focus on variability at different spatio-temporal scales to better comprehend the variables that affect species' ranges and identify threats to vulnerable species. Prompt actions are needed to protect herbaceous and shrub vegetation in this region, highly demanded for agriculture and cattle grazing.

## Introduction

Biodiversity loss is occurring at an unprecedented rate, with many species being lost before they are acknowledged [1]. This loss results from various threats that differentially impact organisms across distinct spatial and temporal scales. Climate change and habitat degradation are particularly influential for ectothermic organisms like reptiles and represent the main causes of biodiversity loss [2–6]. Therefore, an understanding of the determinants of species distribution is essential to identify threats and establish effective short- and long-term conservation and management plans [7–9]. The lack of information about threats to biodiversity

biodiversidad.co/permisos/resource?r=udea_
mhua_2022_atractuslasallei

**Funding:** The author(s) received no specific
funding for this work.

**Competing interests:** The authors have declared
that no competing interests exist.

hinders the optimization of conservation efforts and contributes to an absence of engagement
from local communities in preserving their biotic richness [10]. This situation is particularly
common for tropical snakes, because there is a considerable information gap about their distri-
bution, ecology, and natural history [11, 12]. Threatened snakes are in a state of heightened
vulnerability, not only due to indiscriminate killing resulting from widespread fear among
people but also due to the lack of information about their natural history, which complicates
the assessment of their conservation status [13]. This lack of knowledge is even more apparent
in species with cryptic-fossorial habits, which typically spend a considerable portion of their
lives beneath the surface [14].

With the advent of publicly available databases and remote sensing, several methodologies
have been developed to approximate species distributions at broad scales, with ecological cor-
relative niche modelling being one of the most widely used and effective methods [8, 15, 16].
Niche modelling can be used to predict potential distribution based on the relationship
between geographic (where occurrences are registered) and environmental space (where the
niche is defined) [17], typically using abiotic environmental predictors (climate) to estimate
suitability [18]. However, a significant critique of these models is the lack of information about
species' absences. Presence-only models can take advantage of the copious number of detec-
tions stored in global repositories such as GBIF and iNaturalist but are conditioned to make
coarse spatio-temporal predictions and are susceptible to sampling biases, that prevent them
from robustly estimating prevalence or probabilities of occurrence [19–22].

Methods to generate presence-nondetection data, which enable a direct estimation of the
species occupancy (proportion of an area occupied by a species) require repeated visits to mul-
tiple localities to estimate absence. Occupancy models simultaneously estimate species occu-
pancy and detectability (i.e., the probability of detecting a species if it present) based on
systematic and repetitive sampling. Occupancy models offer a reliable alternative to estimate
species distribution and habitat use at a fine scale [19, 23, 24]. Researchers must revisit a series
of sites and acquire information during a specific time and spatial intervals. Sampling design
must fulfil a series of assumptions (that can sometimes be relaxed), such as the closure assump-
tion (sites are closed to changes in occupancy status within sampling period) and independent
detection histories. Both approaches can be applied at various scales and have proven useful
for studying the distribution of fossorial-cryptic snakes with conservation goals in mind
[25–27].

*Atractus lasallei* is a fossorial-cryptic Colombian endemic snake, characterized by its small
size, that inhabits humid to very humid pre-montane forests in the northern Central and
Western Andes [28–30] and currently classified as "Least Concern" (LC) by the IUCN. Despite
its common status, its populations are highly fragmented, and there is a presumed continuous
decline in the numbers of adult individuals [29]. This decline is attributed to several threats
that affect Colombia's snake fauna, including indiscriminate killing, habitat alteration, local
climate change, and reduction in prey availability [11, 13]. These threats are of particular con-
cern for *A. lasallei* considering that the region it inhabits has historically suffered substantial
deforestation, mainly for mining, livestock farming and agriculture [31], resulting in a land-
scape predominantly characterized by pastures and cultivated fields [32]. However, to date,
most studies about *A. lasallei* have primarily focused on its taxonomy and systematics, and
there are limited publications addressing ecological aspects such as habitat use [30, 33, 34].
Further, some species occurrences seem to be beyond its reported distribution range.

The fossorial and cryptic habits, coupled with the small distribution range of *A. lasallei*,
make it an ideal model to explore the complementary implementation of ecological niche and
occupancy modelling. We hypothesize that the geographic distribution of *A. lasallei* can be
estimated through approaches implemented at different spatio-temporal scales. While niche

models can make use of all records available but compromising temporal and spatial resolution, occupancy models require data collection under a strict sampling design that often limits the temporal and spatial extent. We take advantage of these trade-offs and predict that niche models will identify environmental requirements of the species at broad scales (e.g., Grinnelian niche), and occupancy models will identify variables more representative of essential interactions for the species (e.g., Eltonian niche). Given its fossorial-cryptic habits, we expect ground-level variables to be more informative for modelling its potential distribution, and a strong association with habitats characterized by low vegetation such as grasslands and shrubs. We estimated the geographic distribution area of *A. lasallei* through the combination of both modelling approaches, emphasizing the applicability of such results for conservation and management purposes.

## Materials and methods

### Ecological niche model and potential distribution

Presence data were acquired from three sources: 1) specimens from biological collections, obtained from the Global Biodiversity Information Facility (accessed 22 March 2022) [35] and most of them revised in situ in the following collections: MHUA-Museo de Herpetología Universidad de Antioquia (Curator: J.M. Daza-Rojas), CSJ-h-Museo de Ciencias Naturales de La Salle (D.Z. Urrego), CBUCES-D-Colecciones Biológicas Universidad CES (J.C. Duque); 2) iNaturalist records obtained directly from them (accessed March-May 2022; we did not use iNaturalist records from GBIF) by searching *Atractus* records in the northwest of Colombia that included pictures that allowed verification through morphology (coloration patterns, and scale counts when possible), and 3) individuals encountered during the field phase for occupancy models. Identification of individuals was based on the original species description and taxonomic revisions of the genus [28, 33]. Further, a geographical filter was applied to presence records that were within 1 km of each other to reduce spatial autocorrelation [36, 37].

We used the final database to delimit the species accessible area or M [38] based on the intersection between the minimum convex polygon generated with 50 km buffers around each presence record using QGIS v.3.10 [39], and the biogeographic regions of the northern Central Cordillera and the Western Cordillera of Colombia [40] (Fig 1).

The environmental variables for the niche model included topographic variables, atmospheric climate variables including temperature corrected to ground level [41] and soil variables [42] (S1 Table). Climate variables represent long-term averages (1980–2010 in the case of atmospheric variables, and 2000–2020 in the case of ground-level temperature; see S1 Table). These variables were selected based on previous research findings regarding the distribution of semi-fossorial reptiles [25, 43–45]. All variables were used in the models at a spatial resolution of 1 km. Variables with finer resolutions were resampled using the bilinear method, with the R-package "raster" v.3.5 [46] in R v.4.2.1 [47]. Subsequently, a Spearman correlation test (S1 File) was conducted to select non-correlated variables ($< 0.8$), using R-package "corrplot" v.0.92 [48]. Finally, two sets of variables were created, one that included two ground-level temperature variables estimated at five centimeters above the ground (S1 Table) [41], and the second included the same two variables but measured at atmospheric level [49]. Models were calibrated with each data set independently, ensuring all variables used were not correlated.

The ecological niche model was generated using the maximum entropy algorithm [15] through the R-package "kuenm" [50]. This methodology allows the evaluation of different sets of environmental variables (set 1 and set 2, S1 Table) and various model parameterizations to ultimately identify the best model according to a set of criteria. We allowed the regularization parameter to vary from 0.1 (very strict in relation to observed values) to 5 (more flexible in

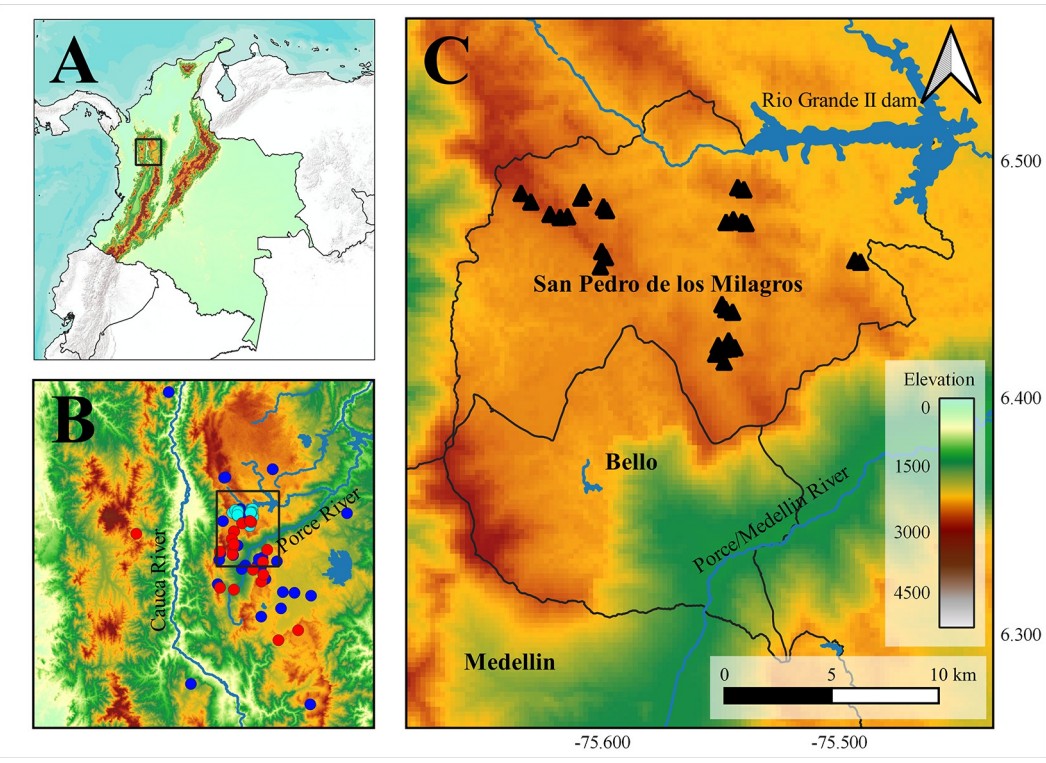

**Fig 1. Geographical location of study area.** (A) Study Area encompassing northern Western and Central Cordilleras. (B) Presence records of *A. lasallei*, where light blue dots depict specimens collected during the field phase of occupancy models, red dots represent iNaturalist records, and dark blue dots indicate specimens in biological collections. (C) Transects represented as black triangles. Green-Brown color-scale represents altitude. Reprinted from [56] and OpenStreetMap under a CC BY license, with permission from [56] and ©OpenStreetMap contributors, original copyright 2020 and 2024, respectively.

relation to observed values), where 1 is the default value. We also evaluated across linear (l), quadratic (q), and linear-quadratic (lq) responses. The models were trained using a 20% random partition of the occurrence data for model evaluation. The evaluation criteria included omission rate (<5%), partial receiver operating characteristic (partial ROC), area under the curve (AUC ratio>1), and the Akaike Information Criterion corrected for sample size (AICc) [51]. In case several models achieved the evaluation criteria, we performed a consensus model using the median of the selected models. Finally, to obtain the geographic projection, a cutoff threshold was applied using the 10 percentile training presence criteria from the best model(s) to generate a presence/absence map.

## Occupancy models

To identify fine-scale factors influencing the occupancy of *A. lasallei* within its known distribution area, single-season occupancy models were employed [24]. The sampling design followed the recommendations of a previous study for semi-fossorial species [52], wherein 30 linear transects of 100 m x 2 m were established within the sampling area, spaced at a minimum of 200 m apart to ensure independence of detection histories across sites (Fig 1). Each transect was equipped with nine artificial cover objects (three roof tiles, three boards, and three plastic sheets), which were installed a minimum of two months prior to sampling for the organisms to habituate to their presence (S1 Fig). The transects were surveyed between

October 2021 and January 2022 to ensure consistent occupancy status during the sampling period (closed-site assumption) between 8 AM and 4 PM. Surveys involved searching beneath leaf litter and under cover objects (both artificial and natural). Each transect was surveyed a minimum of four times, with visits spaced at least two weeks apart to satisfy the assumption of temporal independence. Animals were photographed and examined in the field to ensure correct identification (Approval Act No. 138, February 9, 2021, granted by the Committee on Ethics for Animal Experimentation, Universidad de Antioquia).

Occupancy models were constructed using the R-package "unmarked" v.1.2.5 [53] implemented in the R software. All covariates were standardized (mean = 0, units in standard deviations) prior to modelling. To identify the best models, we first established the best detection model assuming constant occupancy, and then we used this detection model in all occupancy models [54]. To model detectability, we included as covariates, the number of cover objects, both natural and artificial (N_obj); vegetation height (Veg_H) [55]; soil moisture (Soil_moisture); and soil temperature (T_ground), both measured using a HOBO proV2 datalogger beneath a roof tile or under the object where an individual of the species was located at the time of each visit (vegetation height was not related to the number of cover objects, S5 Fig). As covariates for occupancy, we used vegetation height (Veg_H) [55]; terrain slope (Slope); topographic convergence (Con); compound topographic index (CTI) [56]; annual mean soil temperature (Tprom), maximum temperature of the warmest month (Tmax), and minimum temperature of the coldest month (Tmin) [41]; depth of leaf litter (Leaf_Dep) and depth of the 0 horizon (Hori0), both measured in the field using a soil auger; euclidean distance to the nearest house (D_house), nearest forest (D_forest), and nearest water body (D_water). These distances were estimated in QGIS [39], identifying the nearest houses and forests to the centroid of each transect using satellite imagery from GoogleEarth (https://www.google.com/intl/es/earth/). To calculate the distance to water bodies, it was necessary to construct a detailed hydrographic network for the area using a 12.5 m resolution DEM obtained from Alaska vertex (https://search.asf.alaska.edu/), utilizing the hydrology toolbox in ArcGIS Pro v.2.7 [57].

A total of 87 biologically plausible and simple models were evaluated, each including one or two variables (S2 Table), 20 of the models were for the detection component with constant occupancy, and the remaining models were for the occupancy component. Finally, to evaluate model fit to our data, we performed a parametric bootstrap test on the chosen model, using the *parboot* function of R package "unmarked" v.1.4.1 [53]. This test generates multiple sets of data iteratively from the best model and then compares these sets with the detection histories obtained in the field. A chi-squared test was employed to evaluate the null hypothesis that the observations are consistent with the proposed model.

## Integration of models

To estimate the species' realized distribution area [58], we used the binary (presence-absence) geographic projection from the consensus niche model to identify the areas where the macro conditions were suitable and applied the best occupancy model within those areas at a higher spatial resolution ($0.00025° \approx 27$ meters). Finally, the resulting map was transformed into a binary outcome using a threshold of 0.78, based on the Q3 (third quartile) of the occupancy distribution values of that map; this threshold corresponds to 4 m of vegetation height according to the best occupancy model (Fig 2), which is biologically justified if we consider that all presence records obtained in the field phase were found in places with vegetation below 4 m.

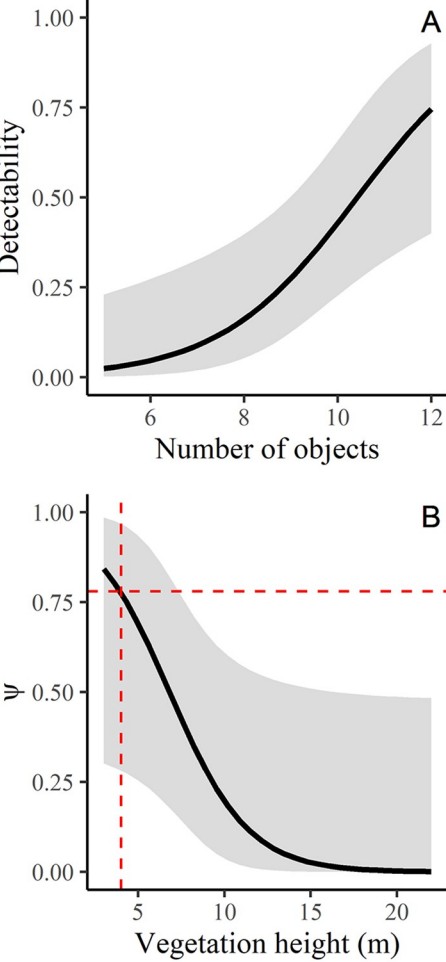

**Fig 2. Occupancy modeling results.** (A) Relationship between the number of cover objects and detectability. (B) Occupancy as a function of vegetation height measured in meters (m), where dashed red lines represent the 4 m vegetation height and its expected occupancy value. Both graphs constructed from the best model.

## Results

### Ecological niche model and potential distribution

A total of 164 records were obtained, out of which 83 referred to specimens from biological collections. Two of these records were collected dead in the northern Western Cordillera, representing a new locality (collection permit under resolution 0524 by ANLA, https://ipt. biodiversidad.co/permisos/resource?r=udea_mhua_2022_atractuslasallei). Additionally, 25 records were harnessed from iNaturalist (S3 Table), and 56 individuals were encountered pre and during field sampling. Following the removal of spatially aggregated records, a dataset of 54 occurrence records for niche modelling was obtained.

We found four ecological niche models that met the established evaluation criteria. These models shared the same feature classes and set of variables but had slight variation in the magnitude of the regularization multiplier, thus a consensus model was created using the median of the selected models. These models included atmospheric level temperature variables (Table 1). The most informative variables for the best models included slope, average annual

**Table 1. Selected models that met evaluation criteria: AUC ratio, probability value (Pval) of partial ROC (pROC), omission rate and AICc.** These models were used to create the consensus model. Model names follow this structure separated by underscores: Model (M), Regularization multiplier (0.5–5), Feature classes (F_l, F_q, and F_lq), and set of variables used (set_1 or set_2).

| Model | Mean AUC ratio | Pval pROC | Omission rate (%) | AICc | Delta AICc | W AICc | Num parameters |
|---|---|---|---|---|---|---|---|
| M_0.5_F_lq_set_2 | 1.787 | 0 | 0 | 935.851 | 0 | 0.0172 | 10 |
| M_0.6_F_lq_set_2 | 1.791 | 0 | 0 | 936.684 | 0.833 | 0.0125 | 10 |
| M_0.3_F_lq_set_2 | 1.777 | 0 | 0 | 937.189 | 1.338 | 0.0074 | 12 |
| M_0.7_F_lq_set_2 | 1.795 | 0 | 0 | 937.527 | 1.675 | 0.0091 | 10 |

precipitation, evapotranspiration, and maximum temperature of the warmest month. The response curves for these variables indicated that the species is associated with flatter slopes (<10˚), relatively high average annual precipitation (200–3500 mm), mean evapotranspiration values between 80 and 120 mm per month, and maximum temperatures between 17 and 22˚C (S2 Fig).

The predicted potential distribution area for the species covers 6182 km$^2$ after applying the 0.096 threshold, obtained from averaging the 10 percentile training presence thresholds from the best models (Table 1), to generate a presence/absence map. The geographical projection reveals three primaries' zones of distribution, including the northern Central Cordillera and two smaller centres along the Western Cordillera, supporting the few occurrence records of the species in this mountain range (Figs 1 and 3).

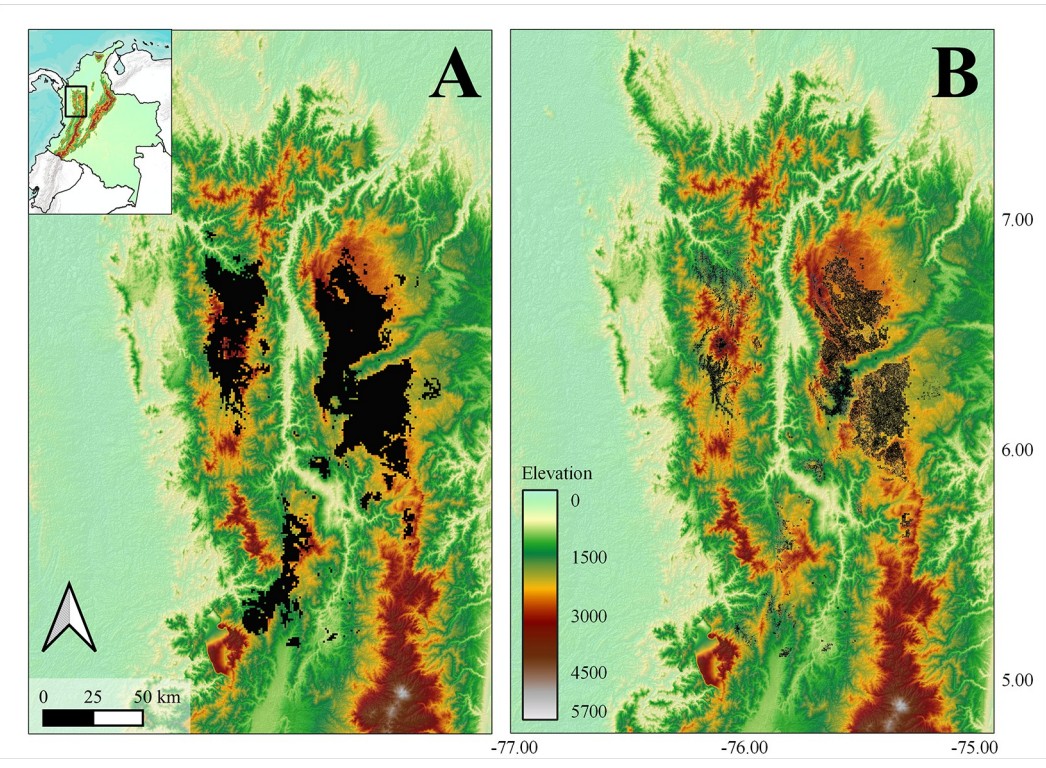

**Fig 3. Potential and realized distribution models.** Comparison between (A) potential and (B) realized distribution area estimated from niche modelling and the integration of niche and occupancy models respectively. Green-Brown color-scale represents altitude. Reprinted from [56] under a CC BY license, with permission from [56], original copyright 2020.

**Table 2. Statistical comparison of detectability submodels using the Akaike information criterion corrected for sample size.**

| Model | K | AICc | Delta AICc | AICcWt | Cum.Wt |
|---|---|---|---|---|---|
| p(N_obj) ψ(.)* | 3 | 76.753 | 0.000 | 0.714 | 0.714 |
| p(Veg_H) ψ(.) | 3 | 79.696 | 2.943 | 0.164 | 0.878 |
| p(.)ψ(.) | 2 | 81.658 | 4.905 | 0.061 | 0.939 |
| p(T_ground) ψ(.) | 3 | 82.961 | 6.207 | 0.032 | 0.971 |
| p(Soil_moisture) ψ(.) | 3 | 83.163 | 6.410 | 0.029 | 1.000 |

Best submodel (*). Models' names show variables used in detectability submodel (p) and occupation submodel (ψ). N_obj: Number of cover objects, Veg_H: Vegetation height, T_ground: ground temperature, Soil_moisture: moisture of ground.

## Occupancy models

Out of 145 surveys (30 sites x 5 repetitions, minus 5 surveys not conducted), *A. lasallei* individuals were recorded on 14 occasions (9.6%) at 7 sites (23% naïve occupancy) (S3 Fig).

The variable that best explained detection was the number of cover objects, which considers both natural and artificial objects (Table 2). The best occupancy model (deltaAICc < 2) only included vegetation height as a covariate (Table 3). Further, the parametric bootstrap test indicated that the observations fit well with this simple occupancy model (p-value = 0.119).

## Integration of models

The estimated realized distribution area was 935.3 km$^2$ encompassing the northern part of the Central Cordillera and a few areas in the Western Cordillera (Fig 3). This area represents 35.7% of the potential distribution area.

**Table 3. Statistical comparison of the best occupancy models.** The inclusion of vegetation height can be observed in the models with lower AICc.

| Model | K | AICc | Delta_AICc | AICcWt | Cum.Wt |
|---|---|---|---|---|---|
| p(N_obj) ψ(Veg_H) * | 4 | 70.090 | 0.000 | 0.324 | 0.324 |
| p(N_obj) ψ(Veg_H+Hori0) | 5 | 72.517 | 2.428 | 0.096 | 0.420 |
| p(N_obj) ψ(Veg_H+Slope) | 5 | 72.536 | 2.446 | 0.095 | 0.516 |
| p(N_obj) ψ(Veg_H+Tmax) | 5 | 72.615 | 2.525 | 0.092 | 0.608 |
| p(N_obj) ψ(Veg_H+D_house) | 5 | 72.619 | 2.530 | 0.091 | 0.699 |
| p(N_obj) ψ(Veg_H+CTI) | 5 | 72.647 | 2.557 | 0.090 | 0.789 |
| p(N_obj) ψ(Veg_H+D_forest) | 5 | 72.906 | 2.817 | 0.079 | 0.869 |
| p(N_obj) ψ(Veg_H+Con) | 5 | 72.990 | 2.900 | 0.076 | 0.945 |
| p(N_obj) ψ(Slope) | 4 | 76.321 | 6.231 | 0.014 | 0.959 |
| p(N_obj) ψ(Con) | 4 | 76.462 | 6.372 | 0.013 | 0.972 |
| p(N_obj) ψ(Tmax) | 4 | 77.409 | 7.320 | 0.008 | 0.981 |
| p(N_obj) ψ(D_water) | 4 | 77.867 | 7.777 | 0.007 | 0.987 |
| p(N_obj) ψ(D_house) | 4 | 78.269 | 8.179 | 0.005 | 0.993 |
| p(N_obj) ψ(Leaf_Dep) | 4 | 78.803 | 8.713 | 0.004 | 0.997 |
| p(N_obj) ψ(Hori0) | 4 | 79.395 | 9.305 | 0.003 | 1.000 |
| p(N_obj) ψ(Tmax+Hori0+Slope+D_house+CTI+D_forest) | 9 | 89.356 | 19.266 | 0.000 | 1.000 |

Best model (*), detectability (p), occupancy (ψ). N_obj: Number of cover objects, Veg_H: Vegetation height, Hori0: Depth of the zero horizon, Slope: Terrain slope, Tmax: Maximum temperature of the warmest month (ground-level), D_house: Distance to the nearest house, CTI: Compound Topographic Index, D_forest: Distance to nearest forest, D_water: Distance to nearest bodies of water, Con: Topographic convergence, Leaf_Dep: Leaf litter depth.

## Discussion

We provide the first estimates of the geographic distribution of *A. lasallei*, whose ecology, including climatic and habitat requirements, were poorly known. The potential geographic distribution of this species was expanded to include the northern section of the Western Cordillera in Colombia, where new records were obtained during this project. Our results indicate that this snake is associated with flat terrain (<10˚), characterized by cool weather (maximum temperatures below 27˚ C; S4 Fig) and high annual precipitation (between 2500–3500 mm). Occupancy models supported an association with habitats characterized by low vegetation cover and an increase in detection related to the number of artificial or natural cover objects. The integration of niche and occupancy models generated an estimate of 935.3 km$^2$ as the actual distribution area for this species (Fig 3). This is well below the mean of the distribution of snake geographic ranges [59, 60] and low enough to consider the species as vulnerable according to IUCN guidelines [61].

The variables identified as important in the niche and occupancy models reveal interesting aspects of the species' ecology, but also provide a means to highlight current threats. The best potential distribution model identified slope as the most informative variable. The species is associated with sites characterized by low slopes (< 10˚) that in this region are highly transformed for cattle grazing and agriculture. Moreover, we found that atmospheric temperature variables were better predictors than ground-level corrected variables, which was unexpected for a semi-fossorial organism that experiences temperature at the level of its microhabitat. Ground-level variables were highly correlated with atmospheric maximum temperature (0.92 Pearson correlation), thus, there seemed to be no large differences between both [41]. Ground level variables should be more informative at finer spatial and temporal scales, which unfortunately are not currently available for the area of interest. We suggest exploring alternative sources of microclimatic data [62, 63].

Vegetation height best explained the species' occupancy pattern, supporting the hypothesized strong association of *A. lasallei* with habitats characterized by low vegetation such as grasslands and shrubs (Fig 2). We hypothesized that the transformation of vegetation in most of the northern central Andes by gold mining and further introduction and propagation of grasses such as *Pennisetum clandestinum* for livestock farming, could represent a trade-off for *A. lasallei* because prey availability is higher in those places where the snakes are more likely to be killed by humans [64]. Changes in vegetation cover may benefit some reptiles due to better thermoregulation conditions [65], or an increase in prey availability because grasslands are often enriched with organic matter providing favourable conditions for earthworms [66, 67], the main prey of *Atractus* snakes [43, 68, 69]. Given that the distribution of *A. lasallei* is dominated by grasslands [32], we expect frequent encounters between these snakes and rural people, often resulting in the death of snakes. Indeed, conservation efforts should be concentrated on environmental education programs targeted towards local communities.

The previous results from the niche and occupancy models support two ideas well established in ecology in relation to the observed geographic distribution of species. One is that the distribution is the outcome of a hierarchy of environmental filters [70–72] that may act at different spatio-temporal scales. A second idea is that variables that determine this distribution include requirements and impacts (static and dynamically linked variables) that may also act at different spatiotemporal scales. Selection at broader geographic scales is likely driven by physiological requirements of individuals to certain abiotic conditions, such as terrain relief and climate. In contrast, selection at finer scales might result from individual experiences and learning, leading to preferences for more dynamic niche variables such as prey and predator density. Even though we only used static variables in both models, we believe vegetation height

might be thought as a proxy for the potential interactions at a site. It is also important to note that biotic variables such as vegetation height were only used in the occupancy models due to its relatively high temporal variability.

The inclusion of the northern section of the Western Cordillera in the geographic range of *A. lasallei* was confirmed based on the identification of various specimens (see lepidosis, S4 Table). This biogeographic pattern is consistent with that described in some disparate taxa such as plants and birds, suggesting a close relationship between the biotas of the northern Central and Western Cordilleras [40, 73, 74]. This pattern highlights the need for phylogenetic studies to clarify the relationship between the *Atractus* lineages in the Andean region and the continuous exploration of the complicated geography of the Andes.

One strategy to improve future explorations is to maximize our detection ability. Occupancy models revealed that the number of cover objects is directly related to detectability. This aligns well with findings from other snakes [52] and results for salamanders in the Great Smoky Mountains National Park, where a similar methodology identified detection varying with the sampling method [75]. However, it's worth mentioning that out of the 14 specimens encountered during field sampling, only two were found under artificial cover objects—an adobe tile and a wooden board—placed in transects with low secondary vegetation and few natural cover objects. This could be explained by: (1) the relatively small size of the objects used, which can be associated to less thermoregulation opportunities [65] and (2) the time that the objects were left in place before data collection began, since snakes often take time to become accustomed to such objects [52, 76]. According to our findings and field observations, we suggest that future studies that focus on fossorial reptiles should consider the size, thickness, and material of the cover objects to improve the species´ thermoregulation opportunities and the detection probabilities.

Our results provide a reference to compare with the two geographic distribution methods used by the IUCN, the Extent of Occurrence (EOO) and the Area of Occupancy (AOO), that represent different aspects of geographic range size [61]. EOO measures the overall geographic spread of the localities at which a species is found, while AOO tends to be much smaller, usually a small fraction of the EOO, and is heavily dependent on the spatial resolution of both the distribution data and the method of measurement employed [59]. We used the same species presence records used for the niche models to estimate EOO and AOO through the R package "ConR" v.1.3.0 [77] (using default parameters) and compared both range sizes with our estimated realized distribution area. For IUCN criteria we obtained an area of 9383 km$^2$ (EOO) and 200 km$^2$ (AOO), while our model estimate of the realized species distribution area was between the two other estimates (935.3 km$^2$). We consider this represents a better approach when trying to assign a conservation category.

Following our results, *A. lasallei* could be considered Vulnerable, not only because the distribution area based on IUCN threshold for Extent Of Occurrence and Area Of Occupancy (20000 km$^2$ and 2000 km$^2$ respectively) [61], but because their populations are highly restricted, fragmented, and with presumed decreasing tendencies [29]. However, these considerations and the ongoing growth of the human population and fast land transformation in this region could place *A. lasallei* at an even higher threat category.

## Supporting information

**S1 Table. Environmental variables for the niche models.**
(DOCX)

**S2 Table. Models tested in the occupancy modeling process.**
(DOCX)

**S3 Table. *A. lasallei* voucher and iNaturalist observations used for niche modelling.**
(DOCX)

**S4 Table. Specimen scale count form Western Cordillera.**
(DOCX)

**S1 File. Spearman correlation test for niche modeling variable selection.**
(DOCX)

**S1 Fig. Artificial cover objects during the sample phase for occupancy models.**
(DOCX)

**S2 Fig. Response curves of most important variables from selected niche model.**
(DOCX)

**S3 Fig. Detection history from sampling phase.**
(DOCX)

**S4 Fig. Elevation distribution of *A. lasallei*.**
(DOCX)

**S5 Fig. Relationship between vegetation height and the number of cover objects.**
(DOCX)

## Acknowledgments

This work wouldn't have been possible without the selfless assistance of the local residents where the field phase was conducted, the tremendous patience and support of our family and friends, the collaboration of numerous researchers and biological collections, the relevant insights from professors, peers, and, of course, the realm of boundless learning that the Universidad de Antioquia provided, fostering the development of this project, and providing the capture permit (collection permit under resolution 0524 by ANLA, https://ipt.biodiversidad.co/permisos/resource?r=udea_mhua_2022_atractuslasallei). The authors are very grateful to Vivian P. Páez, Juan C. Arredondo and Jorge H. Valdez for her initial guidance, to Juan M. Daza for his comments, the editor Mark A. Davis and two anonymous referees, as well Brian J. Halstead for their critical reading, comments and suggestions in earlier versions of this work. a special mention to the GHA and EcoEvo colleagues for their selfless help.

## Author Contributions

**Conceptualization:** Camilo Alejandro Cruz-Arroyave, Felipe A. Toro-Cardona, Juan Luis Parra.

**Data curation:** Camilo Alejandro Cruz-Arroyave.

**Formal analysis:** Camilo Alejandro Cruz-Arroyave, Felipe A. Toro-Cardona, Juan Luis Parra.

**Investigation:** Camilo Alejandro Cruz-Arroyave, Felipe A. Toro-Cardona, Juan Luis Parra.

**Methodology:** Camilo Alejandro Cruz-Arroyave, Felipe A. Toro-Cardona, Juan Luis Parra.

**Resources:** Camilo Alejandro Cruz-Arroyave, Felipe A. Toro-Cardona, Juan Luis Parra.

**Software:** Camilo Alejandro Cruz-Arroyave, Felipe A. Toro-Cardona, Juan Luis Parra.

**Supervision:** Camilo Alejandro Cruz-Arroyave, Felipe A. Toro-Cardona, Juan Luis Parra.

**Validation:** Camilo Alejandro Cruz-Arroyave, Felipe A. Toro-Cardona, Juan Luis Parra.

**Visualization:** Camilo Alejandro Cruz-Arroyave, Felipe A. Toro-Cardona, Juan Luis Parra.

**Writing – original draft:** Camilo Alejandro Cruz-Arroyave, Felipe A. Toro-Cardona, Juan Luis Parra.

**Writing – review & editing:** Camilo Alejandro Cruz-Arroyave, Felipe A. Toro-Cardona, Juan Luis Parra.

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
