## [Decision Letter · Decision Letter 0]

1 May 2024

PONE-D-24-04106Integrating niche and occupancy models to infer the distribution of an endemic fossorial snake (Atractus lasallei)PLOS ONE

Dear Dr. Parra,

Thank you for submitting your manuscript to PLOS ONE. After careful consideration, we feel that it has merit but does not fully meet PLOS ONE’s publication criteria as it currently stands. Therefore, we invite you to submit a revised version of the manuscript that addresses the points raised during the review process.

We look forward to receiving your revised manuscript.

Kind regards,

Mark A. Davis, Ph.D.

Academic Editor

PLOS ONE

3. We note that Figure S3 in your submission contain copyrighted images. All PLOS content is published under the Creative Commons Attribution License (CC BY 4.0), which means that the manuscript, images, and Supporting Information files will be freely available online, and any third party is permitted to access, download, copy, distribute, and use these materials in any way, even commercially, with proper attribution. For more information, see our copyright guidelines: http://journals.plos.org/plosone/s/licenses-and-copyright.

a. You may seek permission from the original copyright holder of Figure S3 to publish the content specifically under the CC BY 4.0 license. 

Additional Editor Comments:

After numerous reviewers declined to review this manuscript, a trio of scientists with relevant research experience took up the task of reviewing this manuscript. All three reviewers agree, and I concur, that this manuscript is well written, the analyses well executed, and the overall topic and aims are of substantial value to broader scientific community. I am therefore pleased to recommend minor revisions for this manuscript. All three reviewers provide a number of recommendations to improve the manuscript, with one particularly important point being the need to connect some of the important and interesting context in the discussion with the introduction.

Reviewers' comments:

Reviewer's Responses to Questions

**Comments to the Author**

1. Is the manuscript technically sound, and do the data support the conclusions?

Reviewer #1: Yes

Reviewer #2: Yes

Reviewer #3: Yes

2. Has the statistical analysis been performed appropriately and rigorously? 

Reviewer #1: Yes

Reviewer #2: Yes

Reviewer #3: Yes

3. Have the authors made all data underlying the findings in their manuscript fully available?

Reviewer #1: No

Reviewer #2: Yes

Reviewer #3: Yes

4. Is the manuscript presented in an intelligible fashion and written in standard English?

Reviewer #1: Yes

Reviewer #2: Yes

Reviewer #3: Yes

5. Review Comments to the Author

Reviewer #1: This study adds to the literature by integrating both ecological niche and occupancy modelling to determine the large- and small-scale habitat associations of Atractus lasallei. The statistics and evaluation of the species distribution models were well done and well described in the results and discussion section. Additionally, the authors were very clear about how this research could be used in a conservation framework by comparing their results to the IUCN geographic distribution methods. Two of my major concerns are 1) a few places where additional information would make the manuscript clearer and 2) a lack of data following the PLOS data policy. These concerns are further outlined below. However, with the following edits, this manuscript will be a great contribution to the field.

Major comments:

1. Introduction:

a. While the discussion mentions the hierarchy of environmental filters and the association with the Grinnellian and Eltonian niches, the Introduction does not. I feel the addition of this information could help describe why you would integrate both ecological niche and occupancy modelling.

b. Authors mention their hypothesis in the Discussion (Lines 277-278) but do not mention them at all in the introduction – I believe authors could include hypotheses of both types of modelling in the last paragraph of the introduction.

2. Methods:

a. For the Ecological Niche Model and Potential Distribution section, authors should add the dates at which they acquired the information. For example, “… Global Biodiversity Information Facility (accessed XXX),”

b. In Lines 122-123, authors mention that a Spearman correlation test was conducted to select non-correlated variables which is correct. However, in Lines 293-295 authors mention that the ground-level and atmospheric maximum temperatures were highly correlated. Due to their high correlation, they should not have been included in the same models but it is unclear in the Methods as to whether one of them was not included.

c. Additionally, on the atmospheric-level variable and the variable estimated at five centimeters, there is no explanation of how the ground-level temperatures were estimated.

d. For the ecological niche models, authors mention the evaluation of “different sets of environmental variables and various model parameterizations” but do not mention what model parameterizations were tested.

3. Results:

a. While data are provided in S5 about where occurrence records are from (e.g., iNaturalist), to meet the criteria of the PLOS Data policy, Latitude and Longitude should also be included. Additionally, for the PLOS data policy, there should be sufficient data to replicate the study; therefore, shapefiles for the variables in the ecological niche models and data for the occupancy models should also be available.

b. For Table 1. Niche Model Results, the model names are unclear, for example, what is the consensus model, what do the numbers and lq mean?

c. Table 2 and 3: All tables and figures should be able to stand alone without the paper attached therefore, all of the names used in the model names need to be defined in the table description.

4. Discussion: The content of the discussion is good but the structure of a discussion should go in the order of:

a. Mini synopsis

b. Restate main findings

c. Significance and meaning (contextualization)

d. Compare results

e. Impact and applications of research

f. Currently, the discussion does not follow this structure with some of the most important impacts and applications (comparison to the IUCN and potential changes to its listing) in the middle and some of the significance and meaning/ comparing results (Lines 327-342) at the very end.

Minor comments:

1. Line 28: Change “centres” to “centers” for American English, check throughout.

2. Line 31: Add a general/overall conclusion sentence.

3. Line 41: “is” should be “are” because authors are referencing more than one thing.

4. Line 50: I do not believe “conspicuous” is the correct word here; maybe “apparent” instead.

5. Lines 55-56: Cite this and add to references

6. Lines 58-60: Niche modelling is more about the relationship between species occurrence data and environmental data – I am not sure what you mean by “geographic”

7. Line 69: Missing the word scale after fine

8. Lines 74-91: This paragraph could be structured differently to be clearer as to why these models are important for this snake in particular. Personally, I would start with the sentences in Lines 80-91 which would show the need for applying these models and then present niche and occupancy models as a viable option to learn more about its habitat use.

9. Lines 112-115: Figure panel reference should go at the beginning of the sentence e.g., “A) Study area encompassing northern…”. Additionally, what do the green-red colors represent? That should be included in the Figure caption and a scale/legend should be included in the Figure.

10. Line 113: Capitalize “Light”

11. Line 122: What raster package was used? Further, at the first mention of a program, the version should be included.

12. Lines 122-123: What package was used to complete the Spearman correlation test.

13. Lines 128 and 168: R package names are typically in quotes “kuenm” package. Make sure to check this throughout.

14. Lines 132-133: Acronyms need to be spelled out on first use, e.g., “partial receiver operating characteristic (ROC) curve”. Check for this throughout.

15. Lines 144-145: Spell out numbers below 10 (except for times and measurements). Check the rest of the document.

16. Line 162: What version of QGIS?

17. Lines 176-177: What package did you use to perform the chi-square test?

18. Lines 184 and 187: Consistent with other measurements, put a space between 4 and m. Check throughout for consistency.

19. Line 189-192: Use the same format as Fig 1, with (A) and (B) rather than top and bottom. Make sure to add A and B to the figure itself.

20. Line 202: Add ecological before niche to stay consistent.

21. Lines 224-226: See above notes for Fig 1 and Fig 2.

22. Lines 233-235: Starting at “supporting” this statement is more for the discussion not the results.

23. Line 250: Change Atractus lasallei to A. lasallei.

24. Lines 255-256: Can you reword to avoid using “with” three times?

25. Line 257: Change “may provide” to “provided”.

26. Line 261: You should not start a sentence with an acronym so even though you already defined EOO, it should still be spelled out at the beginning of the sentence.

27. Lines 276: Change “even at higher threat category” to “at an even higher threat category”.

28. Line 297: What is NDVI?

29. Lines 308-321: This is a great paragraph!

30. Lines 322-326: I think this paragraph could be included with the previous paragraph since it is only two sentences.

31. Lines 327-342: I think this is a great paragraph and some of this information could be added to the introduction.

Reviewer #2: The manuscript “integrating niche and occupancy models to infer the distribution of an endemic fossorial snake (Atractus lasallei) is a well written and supported research article that meets PlosOne’s publishing criteria. The approach integrating scale into distribution through multiple methods is well done and makes for a compelling read, providing an informed understanding of distribution and occupancy. This paper provides evidence for conservation needs of A. lasellei and frames the conservation in the broader context of IUCN listing criteria and spatial scale. I have provided minor comments to improve on the manuscript, identifying areas for clarification and framing.

General comments:

In several sections “habitat preference” or “prefers” is used; I recommend changing this term to “habitat selection” and “selects” respectively based on Johnson (1980) terminology, where preference requires that resources are equally available.

Johnson, D. H. 1980. The comparison of usage and availability measurements for evaluating resource preference. Ecology 6: 65 – 71. https://doi.org/10.2307/1937156

The discussion brings up different niche concepts (Grinnellian and Eltonian) and extent of occurrence and area of occupancy. These concepts could be mentioned in the introduction to set up the study to strengthen the manuscript.

GBIF and iNaturalist were both used for occurrence data, but GBIF pulls from iNaturalist; authors could clarify if iNaturalist occurrences were all drawn from GBIF, or if their criteria for iNaturalist resulted in additional occurrences. I appreciate the authors including the GBIF DOI to ensure reproducibility and credit.

For the data used in the distribution models, please include years of the occurrence records. Given that some predictors are dynamic (precipitation, temperature), providing a temporal range of the occurrence data used will inform interpretation of the model results. I do appreciate the mention of only using abiotic variables used in the ENM due temporal effects (Line 341-342), but more information could be provided to validate that those abiotic factors are either constant or likely representative during the time the occurrence was observed.

The parameter selection based on ecologically relevant variables is well done (Lines 117-120), and the authors have selected variables relevant to their study species.

For the occupancy models (lines 152-177), I recommend introducing the approach of iteratively running models to determine detectability prior to determining occupancy before discussing the parameters included in the models. On first read, I was concerned that the same parameter was used for detection and occupancy. I will note that the methods are appropriate and well done, so this is just a paragraph organization comment. Leading with the structured analysis approach before mentioning the covariates would strengthen this section.

PlosOne submission guidelines do not have information on keywords; if keywords are allowed, some of the keywords like “Conservation” or “Ecology” seem incredibly broad; authors should consider if there are words that would be better. Methodology terms may be an option to consider such as “maxent” or “single-season occupancy model” so people looking for those frameworks will find the article.

Line comments:

Line 24: I recommend:

“We modeled the potential distribution of A. lasallei based on ecological niche theory [using maxent], and habitat use was characterized while accounting for imperfect detection using a [single-season] occupancy model.”

Line 25: “that” should not be in italics

Line 55-56: The statement that models that account for detection through multiple site visits is more recent than broad scale species distribution models needs a citation, or at least needs more context. Both methods gained strong applications in the 2000s with MacKenzie (2002) introducing methodology for detection probabilities <1 and Phillips et al. (2006) introducing species distribution modeling through a maximum entropy approach. I think reframing this to be how the approaches are complimentary accounting for different spatial scales of distribution and site selection would strengthen the support for your methods and results.

MacKenzie D.I., Nichols J.D., Lachman G.B., Droege S., Royle J.A. et al. (2002). Estimating site occupancy rates when detection probabilities are less than one. Ecology, 83,2248–55. https://doi.org/10.1890/0012-9658(2002)083[2248:ESORWD]2.0.CO;2

Phillips, S. J., Anderson, R. P., & Schapire, R. E. (2006). Maximum entropy modeling of species geographic distributions. Ecological Modeling, 190, 231–259. https://doi.org/10.1016/j.ecolmodel.2005.03.026

Line 74: I recommend changing “viable” to “complimentary”

Line 106: 1 km is a reasonable spatial filtering distance; is there a citation that could be included on A. lasallei dispersal distance that would further support this spatial filtering and provide an ecological context for selecting 1 km? If not, 1 km as stated is sufficient.

Line 132: for the partial ROC analysis, if done in the same keunm R package, please include a citation to Peterson et al. (2008) to clarify methodology.

Peterson, A. T., Papeş, M., & Soberón, J. (2008). Rethinking receiver operating characteristic analysis applications in ecological niche modeling. Ecological Modelling, 213, 63–72. https://doi.org/10.1016/j.ecolmodel.2007.11.008

Line 159: in-line citation is incorrectly formatted and has “(Lembrechts et al., 2021)[40]).” Please remove “(Lembrechts et al., 2021)” and just keep the numeric reference.

Line 219: clarify where the threshold 0.096 came from.

Line 286: change to “resulting in the death of [snakes].” as there could be ambiguity if the human-snake conflict results in the death of people or snakes.

Line 289: “The species prefers sites with low slopes (< 5 degrees);” in the results (Line 209-2010), it’s stated that the species prefer slopes below 20 degrees. From the ENM response curves (S6), it appears that there is selection for low slopes with slopes of 0 degrees being optimal. Check these results and clarify for consistency.

Line 309: change “canopy height” to “vegetation height” for consistency

Tables and Figures

Table 1: spell out the model or identify what parameters are in the model as model names “M_0.5_F_lq_set_2” are not informative to a reader. Write out and define Pval and pROC in the caption.

Reviewer #3: General Comments:

This manuscript evaluates the distribution of a cryptic snake, Atractus lasallei, using two complementary techniques, the Maxent algorithm and occupancy modeling. The authors use the former to evaluate the potential distribution (ecological niche) of A. lasallei based on presence-only records and relatively large spatial scale predictors, and the latter for targeted surveys of transects to identify fine-scale associations of A. lasallei occurrence while correcting for imperfect detection. The authors identified slope, annual precipitation, evapotranspiration, and maximum temperature as important variables defining the ecological niche of A. lasallei, whereas occurrence at specific sites within these large-scale constraints was associated with lower vegetation height (e.g., grasslands and shrublands). The authors also identified that the availability of cover objects was positively associated with detection probability in the occupancy models. The authors compare their results to the IUCN concepts of Extent of Occurrence (EOO) and Area of Occupancy (AOO), and suggest that A. lasallei could be considered Vulnerable using these criteria. The authors also compare the use of presence-only species distribution models to the Grinnellian (or potential) niche, and occupancy to the Eltonian (or realized) niche. Finally, the authors indicate how occurrence of the snake in flat areas with low vegetation heights increases the likelihood of negative interactions with humans that often result in the death of the snake.

I found the manuscript generally well-written. The Introduction provided good background information on the importance of modeling distributions of species and the strengths and weaknesses of the different approaches used in the manuscript. The Methods were appropriate, and the supplemental information supported the Methods and Results well. The authors presented the Results clearly, and the Discussion nicely put the study into context without straying too far or becoming too speculative.

I have relatively few concerns with the manuscript, and most of these appear in the Specific Comments below. In addition to those comments, there are a few items that, in my opinion, warrant some further consideration:

• The authors suggest that the primary difference between presence-only species distribution models and occupancy models is spatial scale, but in my mind that is not the case. In theory, both methods could be applied at the same spatial scale, it is just that occupancy modeling requires accounting for the detection process, which is a cost in terms of study design and field sampling, but a huge benefit in terms of accounting for sampling bias and providing reliable inference. Therefore, the separation of these two methods by spatial scale is somewhat artificial on theoretical grounds, but does have some merit in terms of implementation in the field. I would suggest that occupancy modeling be used in preference to presence-only species distribution modeling as a general guide, but I understand that this is generally not possible over the range of most species.

• The comparison of the two methods to Grinnellian and Eltonian niches is interesting, but here, too, I think the primary difference is more than spatial scale. As I interpret them, the Grinnellian niche is the areas the species could potentially occupy if all biotic constraints were absent (i.e., plenty of resources, competition and predation were not limiting factors, etc.) and the Eltonian niche is where the species occurs after accounting for these biotic interactions. So in my mind, these aren’t necessarily about spatial scale (limiting abiotic factors could change over very small spatial scales for some species), but rather about the variables included in the model (either presence-only SDM or occupancy). In this sense, I wonder if vegetation height is, for these snakes, part of the potential or realized niche. The biotic interactions of the snakes with the vegetation are very limited (i.e., they do not eat the vegetation, it doesn’t compete with them, and it does not eat them), but the vegetation could have a very large effect on the physical environment (temperature, humidity, structural attributes, etc.) that influence the snakes directly and indirectly via their prey and predators (including humans). I suggest either providing more nuance to these concepts and separating them from spatial scale and the type of model used or removing this section of the Discussion.

• Similar arguments apply to the authors’ comparison of their results to the IUCN concepts of EOO and AOO. Here, too, either modeling framework could inform these metrics, though I agree that occupancy modeling is much better-suited to AOO because of its ability to correct estimates of occurrence for sampling bias.

In addition to these general comments, I reference specific comments by line number below.

Specific Comments:

Line 25: Remove italics from “that.”

Line 26: Replace “exhibits preferences for” with “selects.”

Line 48: Delete “the.”

Line 64: I recommend replacing “presence-absence” with “presence-nondetection” to highlight that absence is difficult to ascertain, especially with cryptic snakes.

Line 67: Replace “it’s” with “it is.”

Line 69: Insert “scale” after “fine.”

Lines 70–71: Occupancy models do not require that populations be closed (i.e., no change in abundance because of birth, death, immigration, or emigration), but single-season occupancy models do assume that sites are closed to changes in occupancy status (i.e., empty sites are not colonized and occupied sites do not become extirpated within the sampling period). The other assumptions can be, and in fact often are, relaxed. Applying covariates to occupancy and detection implies that these are not constant, but rather vary with environmental or survey conditions. I suggest rewording this sentence.

Line 88: Delete “up.”

Line 114: Capitalize “light” at the beginning of the sentence.

Line 172: This is a large number of models, especially for 30 transects. Given the exploratory (and predictive, in the sense that the selective model is used for mapping rather than causal inference) nature of the work, however, I think this is OK. It might be worth noting that 20 of the models were for the detection component with constant occupancy, and the remaining models were for the occupancy component.

Line 184: Replace the comma with a semicolon.

Lines 185–187: Yes, but were vegetation height and number of cover objects related at all? The detection probabilities are very low with fewer than 10 cover objects, resulting in a lot of uncertainty in the relationship between vegetation height and occupancy. Given this level of uncertainty, 4 m vegetation height seems like it could exclude a reasonably large proportion of area suitable for the snakes. It would be good to present some indication of the distribution of the number of cover objects and vegetation height among transects either by placing tick marks along the x-axes in Fig. 2 or as histograms (or better, a scatterplot of vegetation height vs. number of cover objects) in a supplemental figure.

Lines 206–207: This belongs in the discussion.

Line 209: Replace “has preference for” with “is associated with.” Preference is generally restricted to mean that given equal availability of resources (usually in a laboratory setting), an animal would choose resource X over resource Y.

Lines 209–212: Some of these response curves have distinct peaks. It might be desirable to narrow the reported range of values associated with snake detection. For example, flatter slopes (< 10°), precipitation between 2000 and 3500 mm, evapotranspiration between 80 and 120, and maximum temperature between 17 and 22 (lower values have a lot of uncertainty) seem to me to be particularly high in the response curves.

Line 218: Remove period from “6182 km2” to avoid confusing readers in some countries.

Line 251: Delete comma following “climatic” and replace “were” with “was.”

Line 253: Replace “prefers” with “is associated with.”

Line 254: Replace “terrains” with “terrain.”

Line 255: Replace “a preference for” with “an association with.”

Line 263: Delete “and” where it precedes “usually.”

Line 265: Delete comma.

Line 276: Replace “even at” with “at an even.”

Line 284: Replace “preys” with “prey.”

Line 289: Replace “prefers” with “is associated with” and delete comma.

Lines 290–297: Ground temperature is a calculated expectation, and I would expect that this would vary substantially on a much smaller than 1 km2 spatial scale that is more relevant to the snakes (in addition to the mechanisms described here). Perhaps that is another reason why it was not particularly predictive. I agree with the sentence on lines 297–298; such microclimatic data are difficult to obtain across larger spatial scales, though.

Lines 315–318: Agreed.

Line 321: Replace “the detection histories” with “detection probabilities.”

Lines 322–326: The recommendation to use a removal design is particularly efficient when species are common, as for A. lasallei in grasslands. Specht et al. (2017) provide particularly useful guidance on alternative occupancy designs under different conditions of ψ and p.

Line 336: Replace “preys” with “prey.”

Fig. 2. I recommend replacing the word “Quantity” in the x-axis label of the top panel with “Number.”

Literature Cited:

Specht, H. M., H. T. Reich, F. Iannarilli, M. R. Edwards, S. P. Stapleton, M. D. Weegman, M. K. Johnson, B. J. Yohannes, and T. W. Arnold. 2017. Occupancy surveys with conditional replicates: An alternative sampling design for rare species. Methods in Ecology and Evolution 8:1725–1734.

6. PLOS authors have the option to publish the peer review history of their article (what does this mean?). If published, this will include your full peer review and any attached files.

Reviewer #1: No

Reviewer #2: No

Reviewer #3: **Yes: **Brian J Halstead

---

## [Author Response · Author response to Decision Letter 0]

25 Jun 2024

Thank you for your constructive feedback on the paper, "Integrating niche and occupancy models to infer the distribution of an endemic fossorial snake (Atractus lasallei)". In order to meet the journal requirements:

1) We revised again the manuscript to ensure it meets PLOS ONE's style requirements, including those for file naming.

2) We included a full ethics statement in the ‘Methods’ section of the manuscript file. This section includes the full name of the ethics committee who approved our sampling protocol.

3) Regarding the figure containing a copyrighted image (Figure S3), the photo belongs to Juan David Martinez Martinez, who confirmed that he has the copyright to the image and grants us permission for its publication under a Creative Commons Attribution 4.0 International (CC BY 4.0) license. We attach a written permission from the copyright holder to publish these figures specifically under the CC BY 4.0 license.

4) We reviewed again our reference list to ensure that it is complete and correct. There are not cited papers that have been retracted.

We extend our gratitude to all Reviewers and have attended all their comments. We modified the manuscript following their suggestions. We paid special attention to the connection between the introduction and discussion, as recommended. We believe the quality of the MS has greatly improved following these changes. 

Please find a detailed answer to each of your comments in the Response to Reviewers document attached.

---

## [Decision Letter · Decision Letter 1]

2 Aug 2024

Integrating niche and occupancy models to infer the distribution of an endemic fossorial snake (Atractus lasallei)

PONE-D-24-04106R1

Dear Dr. Parra,

We’re pleased to inform you that your manuscript has been judged scientifically suitable for publication and will be formally accepted for publication once it meets all outstanding technical requirements.

Kind regards,

Mark A. Davis, Ph.D.

Academic Editor

PLOS ONE

Additional Editor Comments (optional):

Thank you for your consideration of the original reviewer comments and your thoughtful revisions. I look forward to seeing this article in press very soon!

Reviewers' comments:

Reviewer's Responses to Questions

**Comments to the Author**

1. If the authors have adequately addressed your comments raised in a previous round of review and you feel that this manuscript is now acceptable for publication, you may indicate that here to bypass the “Comments to the Author” section, enter your conflict of interest statement in the “Confidential to Editor” section, and submit your "Accept" recommendation.

Reviewer #1: All comments have been addressed

Reviewer #2: All comments have been addressed

2. Is the manuscript technically sound, and do the data support the conclusions?

Reviewer #1: Yes

Reviewer #2: Yes

3. Has the statistical analysis been performed appropriately and rigorously? 

Reviewer #1: Yes

Reviewer #2: Yes

4. Have the authors made all data underlying the findings in their manuscript fully available?

Reviewer #1: Yes

Reviewer #2: Yes

5. Is the manuscript presented in an intelligible fashion and written in standard English?

Reviewer #1: Yes

Reviewer #2: Yes

6. Review Comments to the Author

Reviewer #1: The authors adequately addressed all of my major and minor concerns from the first round of revisions. I have no further recommendations for this article.

Reviewer #2: The authors of “integrating niche and occupancy models to infer the distribution of an endemic fossorial snake (Atractus lasallei)” have addressed all reviewer comments. The manuscript is an enjoyable read that will benefit conservation and demonstrates an advance in understanding conservation needs of A. lasallei. The reviewer responses are good and I believe this manuscript is ready for publication. I have added two requests to the authors; The data is available in the supplementals, dryad materials, and gbif DOI, but I think further transparency would be beneficial.

First, I encourage the authors to include a range of years or some description of sample collection year in the main manuscript rather than leave the data in the supplementals. The CHELSEA datasets are based on 1980-2010, but occurrences range from 1924-2022 (median = 1986, mean = 1985.5, SD = 30.5). I do not think this effects the analyses or interpretation, but I think having some information on the temporality of the samples available to readers in the manuscript rather than solely supplemental 5 would increase transparency.

Second, I would include latitude and longitude along with year in Supplemental 5. Although the dryad files include lat/lon of the samples used in the analysis, the sample IDs are not associated with the location. Having a simplified data set of sample, year, latitude, longitude would be clearer and easier for a reader to follow if attempting to replicate the study.

Despite those two final comments, the authors have conducted a great study and it will be a benefit to applicable conservation of A. lasallei and conservation practices broadly. I greatly enjoyed the opportunity to review this manuscript and I am excited to see it published.

7. PLOS authors have the option to publish the peer review history of their article (what does this mean?). If published, this will include your full peer review and any attached files.

Reviewer #1: No

Reviewer #2: No

---

## [Editor Report · Acceptance letter]

9 Aug 2024

PONE-D-24-04106R1 

PLOS ONE

Dear Dr. Parra, 

I'm pleased to inform you that your manuscript has been deemed suitable for publication in PLOS ONE. Congratulations! Your manuscript is now being handed over to our production team.

Kind regards, 

on behalf of

Dr. Mark A. Davis 

Academic Editor

PLOS ONE